# Acute Effects of Static and Proprioceptive Neuromuscular Facilitation Stretching of the Plantar Flexors on Ankle Range of Motion and Muscle-Tendon Behavior in Children with Spastic Cerebral Palsy—A Randomized Clinical Trial

**DOI:** 10.3390/ijerph191811599

**Published:** 2022-09-15

**Authors:** Annika Kruse, Andreas Habersack, Richard T. Jaspers, Norbert Schrapf, Guido Weide, Martin Svehlik, Markus Tilp

**Affiliations:** 1Department of Biomechanics, Training and Movement Science, Institute of Human Movement Science, Sport and Health, University of Graz, 8010 Graz, Austria; 2Department of Orthopaedics and Trauma, Medical University of Graz, 8036 Graz, Austria; 3Department of Human Movement Science, Faculty of Behavioral and Movement Sciences, Vrije Universiteit Amsterdam, Amsterdam Movement Sciences, 1081 HZ Amsterdam, The Netherlands

**Keywords:** paresis, Achilles tendon, plantar flexors, hold-relax PNF stretching, muscle activity, spasticity

## Abstract

Stretching is considered a clinically effective way to prevent muscle contracture development in children with spastic cerebral palsy (CP). Therefore, in this study, we assessed the effects of a single session of proprioceptive neuromuscular facilitation (PNF) or static stretching (SS) on ankle joint range of motion (RoM) and gastrocnemius muscle-tendon behavior in children with CP. During the SS (*n* = 8), the ankle joint was held in maximum dorsiflexion (30 s). During the PNF stretching (*n* = 10), an isometric contraction (3–5 s) was performed, followed by stretching (~25 s). Ten stretches were applied in total. We collected data via dynamometry, 3D motion capture, 2D ultrasound, and electromyography, before and after the stretching sessions. A mixed ANOVA was used for the statistical analysis. Both ankle RoM and maximum dorsiflexion increased over time (F(1,16) = 7.261, *p* < 0.05, η² = 0.312; and F(1,16) = 4.900, *p* < 0.05, η² = 0.234, respectively), without any difference between groups. An interaction effect (F(1,12) = 4.768, *p* = 0.05, η² = 0.284) was observed for muscle-tendon unit elongation (PNF: −8.8%; SS: +14.6%). These findings suggest a positive acute effect of stretching on ankle function. However, SS acutely increased muscle-tendon unit elongation, while this decreased after PNF stretching, indicating different effects on the spastic muscles. Whether PNF stretching has the potential to cause positive alterations in individuals with CP should be elucidated in future studies.

## 1. Introduction

Cerebral palsy (CP) is a non-progressive neuro-muscular disorder that results from a brain injury occurring before, during, or shortly after birth [1]. Individuals with CP have neurological and musculoskeletal impairments [2], which limit their ability to perform activities and restrict their participation in daily life [3]. Until recently, it has been assumed that muscle weakness and stretch hyperreflexia are the key factors that lead to (fixed) muscle contracture development in individuals with spastic CP [4]. However, growing evidence exists that questions the role of stretch hyperreflexia [5], while emphasizing the importance of the intrinsic pathomorphological features of the spastic muscles.

Previous studies have shown that the muscles of children with spastic CP are structurally and mechanically different than those of typically developing (TD) children [2]. For instance, there is consistent evidence of reduced muscle size (i.e., reductions in muscle volume and length) [6,7,8,9] already present at a very young age in children with spastic CP [10,11], likely resulting from a decrease in sarcomerogenesis coupled with more normal bone growth [2]. Furthermore, low fascicle lengths [12,13,14] have been found in children with CP, compared to TD children, suggesting a lower serial sarcomere number. In addition, spastic muscles show an increased collagen-based connective tissue content [4,15], which may be related to enhanced passive muscle stiffness. These muscular changes restrict the force and maximal muscle power generation capacity, and may contribute to muscle contracture development [4,16,17]. Therefore, a main intervention goal in the therapy of individuals with spastic CP should be to prevent further deterioration of their muscle pathomorphology.

Static stretching (SS) is an important component of physical therapy in individuals with spastic CP. It is an easily and safely applicable, non-invasive method used with the aim to increase muscle belly elongation and length, decrease muscle stiffness, maintain or increase joint range of motion (RoM), and delay the onset of contractures [18]. In general, repeated elongation of the muscle belly during stretching seems to be essential to evoke such changes, as has been shown in animals [19]. Indeed, acutely performed SS has resulted in reduced muscle stiffness in healthy populations [20,21,22]. Nevertheless, there is evidence questioning the overall effectiveness of passive stretching in, for instance, clinical populations with neurological conditions, and especially for the prevention of contractures [23]. However, in individuals with spastic CP, studies have shown the potential of SS to acutely increase ankle joint RoM [24] and improve muscle properties [25]. Nevertheless, following the idea that SS, or other treatments such as splinting or casting, may not be an efficient way to prevent further shortening of spastic muscles, the question remains as to whether or not there is a conservative method that can achieve this goal. Such a method should evoke muscle belly elongation to prevent further shortening of a spastic muscle and the muscle-tendon unit (MTU). Since recent studies in healthy populations (see [26] for review), as well as in patients with paresis, have supported the idea that combinations of stretching with activation might be a promising way to cause positive effects at the muscle-tendon level [27,28,29,30], we assume that proprioceptive neuromuscular facilitation (PNF) stretching might also be an effective approach.

PNF stretching is commonly used with the aim to increase joint RoM and lengthen the MTU [31]. In contrast to SS, PNF stretching usually includes a static contraction of the lengthened target muscle (“contract and relax”), in order to take advantage of autogenic inhibition, which is assumed to assist elongation of the muscle [31]. Moreover, due to the performance of isometric contractions, it can be presumed that long-term PNF stretching may also positively affect muscle strength. Studies in healthy populations have shown greater gains in joint RoM occurring at a faster rate after PNF stretching than SS [32,33,34]. However, to the best of our knowledge, there have been no studies to date that have investigated the effects of PNF stretching on joint function and the muscle-tendon properties of individuals with CP. Consequently, investigating the acute effects of PNF stretching on joint and MTU parameters is important to understand the mode of action of PNF in individuals with CP, and thus to gain knowledge about its potential as a stretching technique for this population.

Therefore, in this study, we investigated the acute effects of PNF stretching, in comparison to SS on ankle joint function, gastrocnemius medialis (GM) muscle-tendon lengthening behavior, and muscle activation in children with spastic CP. We presumed that a single session of PNF stretching would lead to higher increases in ankle joint RoM, maximum dorsiflexion (maxDF), and GM muscle belly elongation, compared to SS.

## 2. Materials and Methods

### 2.1. Recruitment and Randomization

For the randomized clinical trial, which was part of a clinical project (NCT04570358), children with uni- and bilateral spastic CP were recruited from the orthopedic department of the local university hospital. Children were included who were ambulatory, able to accept and follow verbal instructions, and who had no severe contracture defined as maxDF ≥ 0° with knees extended, assessed during clinical examination beforehand. Exclusion criteria were any surgical intervention and/or Botulinum toxin A application in the 12 or 6 months prior to the measurements, respectively.

As a basis for the PNF stretching, we additionally assessed the selective voluntary motor control of all the participants, by use of the “Selective Control Assessment of the Lower Extremity” (SCALE, [35]) tool. The grading was limited to the ankle joint.

For randomization, four groups were created based on both the Gross Motor Function Classification System (GMFCS) level (I + II vs. III) and age (6–10 years vs. 11–15 years). The aim of this stratification process was not to analyze the data according to the four groups, but to produce similar groups. The interventions (SS and PNF stretching) were randomly assigned to a block of four participants in each group. This approach was chosen to balance the groups since the recruitment occurred in a successive manner.

The study was approved by the Ethics Committee of the University of Graz (registration number: 31-130 ex 18/19), and written consent was obtained from each participant in advance.

### 2.2. Data Collection

All measurements were performed by the same experienced assessor on the leg demonstrating the most resistance to ankle dorsiflexion in children with bilateral CP, and on the affected side of the hemiplegic children. During the measurements, the participant lay prone on an examination couch, with the knee placed at ~20° of flexion using a custom-made cushion (Figure 1A), ensuring free sagittal plane ankle movement [36].

Detailed information about the data collection procedure can be found in [37]. In brief, a custom-made footplate was applied to the foot (Figure 1A). This adjustable footplate allowed adjustments to be made to stabilize the subtalar joint, as performed in previous studies [38,39]. Furthermore, an inclino-dynamometer [38] was attached to the footplate (Figure 1B), and reflective markers were placed onto specific (anatomical) sites on the lower leg (Figure 1A). Based on the marker positions captured using a 3D motion capture system (10 cameras, Miqus M3, Qualisys AB, Gothenburg, Sweden), a model of the lower leg was built using Visual3D software (C-Motion, Inc., Germantown MD, USA). Based on the model, the foot sole angle was defined in 3D as the angle between the plane of the footplate (determined by the footplate markers) and the shank (defined by the markers of locations 1 and 2, and the shank cluster) (Figure 1A). Foot sole angles are presented with a 90° offset (i.e., 0° corresponds to the foot sole orientated perpendicular with regard to the shank [39]).

Both directly before and after the stretching interventions (Section 2.3), two dorsiflexion rotations were conducted at a slow speed (totaling four rotations), using the handle of the inclino-dynamometer [37]. During the separate rotations, the foot plate (i.e., foot sole) was moved into dorsiflexion within the sagittal plane, and the foot sole angles and the external torque applied by the examiner, as well as the angular movement velocity, were simultaneously measured [37]. Mean peak rotational velocities applied in the pre-post assessments were 21.6 (3.3)°/s and 22.4 (3.8)°/s in the PNF and SS groups, respectively.

Furthermore, the mechanical lengthening behavior of the GM MTU, the GM muscle belly, and the Achilles tendon (AT, distance from the GM muscle-tendon junction (MTJ) to its insertion onto the calcaneus), as well as muscle activation data, were measured as previously described [37].

To determine the tissue lengthening behavior, a 5-cm linear array ultrasound (US) transducer (LA523, MyLab60, Esaote S.p.A., Genova, Italy) was used in the first step to detect the anatomical landmarks additionally needed for the assessments (Figure 1A): the most superficial point of the medial epicondyle (location 6), the GM MTJ, and the proximal tendinous insertion at the calcaneus (location 3). For the data analysis (Section 2.4), two US images were recorded showing the most superficial point of the condyle and the tendinous insertion point, respectively [37]. Furthermore, reflective markers were placed onto the medial condyle and the calcaneus (Figure 1A). Afterward, a 59-mm linear array transducer (LogicScan 128, Telemed, Vilnius, Lithuania) fitted with a rigid cluster of four reflective markers was fixed over the GM MTJ (Figure 1A) to simultaneously record the displacement of the MTJ during the dorsiflexion rotations [37]. The US images were collected at 60 Hz. This frame rate was chosen in accordance with the other US measurements of the overall clinical project (NCT04570358) that required a faster sampling rate.

The electromyographic (EMG, myon 320, myon AG, Zurich, Switzerland) signals of the tibialis anterior and gastrocnemius lateralis were recorded at 2000 Hz to evaluate the muscle activation in reaction to passive foot sole movement. Skin preparation and surface electrode placement (Blue Sensor N, Ambu A/S, Ballerup, Denmark) were carried out according to the SENIAM guidelines [40] and verified by US.

### 2.3. Stretching Procedures

To investigate the effects of a single stretching session, 10 stretches in total were applied to the plantar flexors, alternating the position of the knee joint (extended/flexed), as displayed in Figure 2.

In the SS group, the ankle joint was moved into maxDF until the participant reported discomfort or joint resistance prevented any further movement. Care was taken to stabilize the subtalar joint during stretching [38,39]. The ankle joint was then held in this position for 30 s, followed by a rest period of 30 s. The procedure was then repeated with the knee in a 90° position (Figure 2). In the PNF group, stretching was performed similarly: the subtalar joint was manually stabilized, and the ankle joint was moved into maxDF forcefully and safely until discomfort was reached [41]. While the foot was kept in maxDF, the participant was asked to perform a submaximal isometric contraction for up to 3–5 s [42] against the investigator’s resistance. The ankle joint was moved further into greater dorsiflexion and held there for the remaining ~25 s, followed by 30 s of rest.

### 2.4. Data Analysis

The data analyses were performed as described in detail in Habersack et al. [37]. Data were captured during two separate dorsiflexion rotations, and mean values were used for the statistical analyses.

Ankle joint parameters (e.g., RoM (i.e., angle range from maximum plantarflexion to maxDF), maxDF) were determined based on the foot sole angle data. Foot sole angle changes were also computed over a common standardized torque interval for all the participants, defined from 0 Nm to 5 Nm externally applied torque. The angle at 0 Nm was defined as the resting ankle joint angle.

The MTJ displacement was manually tracked during the post-processing [43], and the tissue lengthening behavior (i.e., elongation) was analyzed using a previously validated 3D US approach programmed in MATLAB [37] (The MathWorks, Inc., Natick, MA, USA. Absolute MTU length was calculated as the linear distance between the medial epicondyle (i.e., muscle origin) and the AT insertion at the calcaneus. GM muscle belly length was computed as the linear distance between the origin and the MTJ, and tendon length as the distance between the MTJ and its insertion at the calcaneus. For each participant, individual GM muscle belly length changes were calculated over the range from 0 Nm (i.e., resting length) to the maximally applied torque. Furthermore, the tissue elongation was computed over the common torque interval. Passive strain values were also obtained by dividing the muscle and tendon elongation by the tissue resting lengths.

According to previous studies in children with CP [36,44], raw EMG data were filtered with a 6th-order zero-phase Butterworth bandpass filter from 20 to 500 Hz, and the root-mean-square (RMS) envelope of the EMG signal was extracted by applying a low-pass 30-Hz 6th-order zero-phase Butterworth filter on the squared signal. The average RMS-EMG between 10–90% of the foot sole rotation was computed and further expressed as a percentage of the peak RMS-EMG value achieved by the participant during an isometric maximum voluntary contraction measurement (performed in the same experimental session as part of the overall clinical study, NCT04570358). Trials were discarded from the analyses if the average RMS-EMG ≥ 10% of peak RMS-EMG.

### 2.5. Statistics

All the statistical analyses were executed using SPSS (version 26, SPSS Inc., Chicago, IL, USA). The level of significance was set to *p* = 0.05. Data were tested for normal distribution by the use of the Shapiro-Wilk test and distribution histograms. A mixed ANOVA (independent variables: within = time; two levels: pre-stretching, post-stretching; between = stretching groups; two levels: SS, PNF) was used to evaluate the stretching effects. Non-normally distributed data were transformed by use of the Johnson transformation [45].

Due to technical problems, not all data sets were complete (please refer to the result tables). However, no data sets had to be excluded due to exaggerated involuntary muscle activation resulting from velocities applied during rotating the ankle joints.

## 3. Results

### 3.1. Participants

Altogether, 18 children who participated in the overall clinical project (NCT04570358) were recruited for this randomized clinical trial investigating the effects of acute stretching (Table 1).

Since only two children with GMFCS level III could be recruited, who were assigned to the PNF stretching group, the stretching groups differed slightly (Figure 3).

### 3.2. Joint Range of Motion and Muscle Activation

Externally applied peak torque did not differ pre- and post-stretching (Table 2). Furthermore, no significant changes were observed in both maximum plantarflexion and resting foot sole angles. However, a significant time effect was observed for ankle joint RoM and maxDF (Table 2).

Moreover, no significant differences in muscle activation could be found after both acute stretching procedures (Table 2).

### 3.3. Muscle-Tendon Unit, Muscle Belly, and Tendon Length Changes

There was no significant difference (*p* > 0.05, Table 3) in maximum muscle belly length change after both acute stretching procedures. However, a significant time × group interaction effect could be found for MTU elongation (Table 3), determined for the common torque interval, showing increased elongation in the SS group only (SS: +14.6%; PNF: −8.8%).

Time × group interaction for AT length change (Table 3) was not significant (*p* = 0.06), as were the differences in foot sole angle changes (Table 2) and tendon strain (Table 3).

Moreover, no differences could be found for GM muscle belly length changes and GM muscle belly strain calculated over the common torque interval (Table 3).

## 4. Discussion

In this study, we found that one session of either PNF stretching or SS acutely increased maxDF. However, while the elongation of the GM MTU over a common torque interval was reduced after PNF stretching, it increased after SS. Although not statistically significant (*p* = 0.06), this was likely due to a greater elongation of the AT after SS. In contrast to our expectations, the interventions did not lead to increases in muscle belly elongation.

A main intervention goal in the therapy of individuals with spastic CP is to prevent deterioration of their muscular pathology, thus maintaining or even increasing joint function. Consequently, applied treatments should acutely trigger muscle belly lengthening, resulting in an increase in joint RoM. In recent years, PNF stretching has been used to aid the rehabilitation of patients with spasticity or paresis by either facilitating muscle elongation and/or improving muscle strength [31]. Previous studies performed in healthy populations have reported increases in ankle joint dorsiflexion RoM after acute PNF stretching, ranging from 4° up to 6° [20,21,22]. In addition, greater gains in RoM have been found after PNF stretching, compared to SS [32,33,34]. Similarly, increases in joint RoM have also been reported after long-term hold-relax PNF stretching in hemiplegic stroke patients [41]. Therefore, we assumed that acute PNF stretching performed in children with spastic CP would lead to higher increases in ankle joint RoM than SS. However, in contrast to our hypothesis, our study showed that PNF stretching and SS did not cause significantly different changes in ankle joint function (RoM: PNF: +2.2%; SS: +7.4%; maxDF: PNF: +3.3°; SS: +4.1°) over time. Consequently, we rejected our first hypothesis that acute PNF stretching leads to higher increases in ankle joint RoM and maximum dorsiflexion angle than SS. Although not significant, the increases in maxDF were similar to findings of previous studies investigating acute and long-term effects of SS [25,46] and bracing [47]. The reported acute increases may, for instance, positively affect subsequently performed tasks, such as gait. However, the implications of acute increases in maxDF in children with spastic CP still have to be elucidated.

Secondly, we hypothesized that acute PNF stretching would lead to greater GM muscle belly elongation, compared to SS. However, neither acute PNF nor SS affected muscle belly elongation in the present study. This finding is in contrast to observations in healthy subjects demonstrating acute decreases in muscle stiffness after both stretching procedures [20,21,22]. Therefore, we assume that spastic muscles may respond differently to acute stretching than muscles of healthy individuals. Although no studies on the effects of acute PNF stretching in individuals with CP exist, a previous study on SS supported the idea that spastic muscles are indeed able to respond to a stretch stimulus. Theis et al. [25] reported that maxDF as well as maximum GM muscle belly length increased progressively with each stretch (five stretches held for 20 s), and the maximum dorsiflexion angle increased by similar changes in muscle and tendon lengths (~6%) [25]. Moreover, we observed that the stretching techniques had a different effect on the mechanical behavior of the GM MTU, showing increased elongation after SS (+14.6%), but a decrease in elongation after PNF stretching (−8.8%) over a common torque interval. Since muscle belly elongation did not differ, our results indicate that the increase after SS results from an increase in tendon elongation (SS: +49.1%), although the difference between groups was not statistically significant (interaction effect: *p* = 0.06). Our finding supports the assumption by Kalkman et al. [48] that SS (for several weeks) affects the related tendon rather than the spastic muscle belly. We speculate that the different findings between our study and that of Theis et al. [25] might have been caused by the differences in methodology and the participants’ characteristics (e.g., CP phenotype, negative features (e.g., poor selective motor control), treatment history, and muscle pathomorphology).

In contrast to the effects observed after acute SS in the present study, there was no increase, but rather a slight, non-significant decrease in MTU elongation calculated over the common torque interval in the PNF stretching group (−8.8%) after the stretching session. Since muscle activity was slightly reduced not only after SS (−15.2%) but also after PNF stretching (−10.8%, non-significant results), we suggest that the lack of effect on MTU elongation might be explained by the impact of PNF stretching on the mechanical tissue properties. The decrease in MTU elongation was accompanied by non-significant decreases in both muscle belly (−1.1 mm) and tendon (−0.7 mm) elongation, which could be explained by an increase in stiffness. While muscle tissue may increase its stiffness by increasing its muscle tone (i.e., its resting tension) by altered calcium handling or by altering the intrinsic properties of the cytoskeleton and/or the extracellular matrix, an increase in tendinous tissue stiffness (i.e., the stiffening effect of PNF stretching) might be questionable. Our observations contradict previous studies in healthy subjects that have reported decreases in MTU and muscle belly stiffness [21,46], as well as tendon stiffness [21], after acute PNF stretching. Despite the suggestion that muscle and tendinous tissue in individuals with spastic CP might react differently to PNF stretching, the differences in the methodological approaches should also be considered. In the present study, the stretching was performed manually by the examiner. In contrast, the stretching in the other studies with healthy subjects was performed with an isokinetic dynamometer [21,49]. However, similar increases in maxDF can be found in this study (PNF: +3.3°) when compared to others (Kay et al. [20]: +5.3°; Konrad et al. [49]: +1.1°), emphasizing a similar stretching effect. Furthermore, we chose a torque interval in which tissue responses should not be influenced by the stretch tolerance of the participants. Based on these considerations, we are confident that the observed tissue responses in both stretching groups are related to the respective stretching techniques. Nevertheless, to date, this is speculative, and a detailed analysis of PNF stretching in both children with CP and TD children is needed for clarification.

This study does have some limitations that should be considered. We assume that the variability in our sample caused by CP may have prevented some results from becoming statistically significant. Therefore, additional studies are needed with a larger sample size to gain further insights into the effects of PNF stretching in individuals with spastic CP. Moreover, adequately matched study groups were not achieved for several reasons, i.e., only 24 children could be recruited in total, dropouts, and technical issues. Furthermore, we note that the applicability of PNF stretching performed in children with CP may be less effective due to the limited motor control of the children and, in particular, their restricted ability to perform isometric contractions. Regarding this aspect, we additionally assessed the selective motor control of all our participants. Most of the children in the PNF group showed impaired selective motor control, but all the children were able to perform the requested tasks. Moreover, it was shown that, when using submaximal contractions, PNF stretching may be just as beneficial as when performed with maximal contractions [42]. Hence, we are confident that the participants could perform the PNF stretching. It should further be noted that we only assessed the GM MTU. Thus, the presented results cannot be transferred to other muscles and/or muscle groups. Finally, we note that the study design was chosen to compare the acute effects of the two stretching techniques, to gain insights into the modes of action. To verify the significance of acute stretching in the target population, inclusion of a control group would have been needed. Such a comparison should be performed in future studies.

## 5. Conclusions

This is the first study to have investigated the acute effects of PNF stretching compared to SS on ankle joint function and lengthening behavior of the GM MTU in children with spastic CP. Comparing the acute effects of the stretching techniques, we observed different behaviors of the GM MTU after the stretch applications, with decreased MTU lengthening after PNF stretching and increased lengthening after SS. Neither stretching procedures affected the mechanical behavior of the GM muscle belly. Our findings are in contrast with those in healthy populations, where an increase in muscle lengthening (SS/PNF stretching) and tendon lengthening (PNF stretching) has been reported. Since PNF stretching is a treatment approach applied in clinical practice, future examinations of the effects of acute and long-term PNF stretching are needed to better understand its potential to cause adaptations at the muscle-tendon level.

## Figures and Tables

**Figure 1 ijerph-19-11599-f001:**
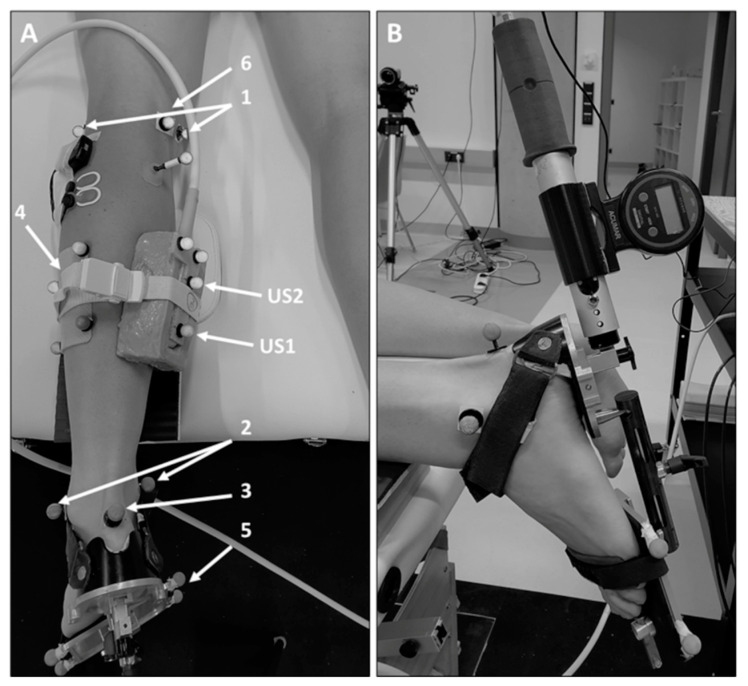
Measurement set-up. (**A**) The image shows the reflective markers, the ultrasound transducer, and the electromyographic sensors that were used for the assessment of the foot sole angles, gastrocnemius medialis muscle belly behavior, and muscle activity throughout the dorsiflexion rotations. Marker placement locations: 1, medial and lateral condyle; 2, medial and lateral malleolus; 3, proximal insertion of the Achilles tendon onto the calcaneus; 4, four marker cluster; 5, four markers attached to the footplate; 6, most superficial point of the medial condyle; US1 and US2, markers placed on the ultrasound probe. (**B**) Presentation of the inclino-dynamometer attached to the custom-made footplate used to perform the dorsiflexion rotations.

**Figure 2 ijerph-19-11599-f002:**
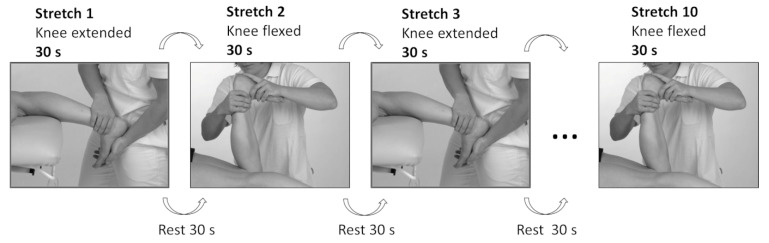
Stretching procedure.

**Figure 3 ijerph-19-11599-f003:**
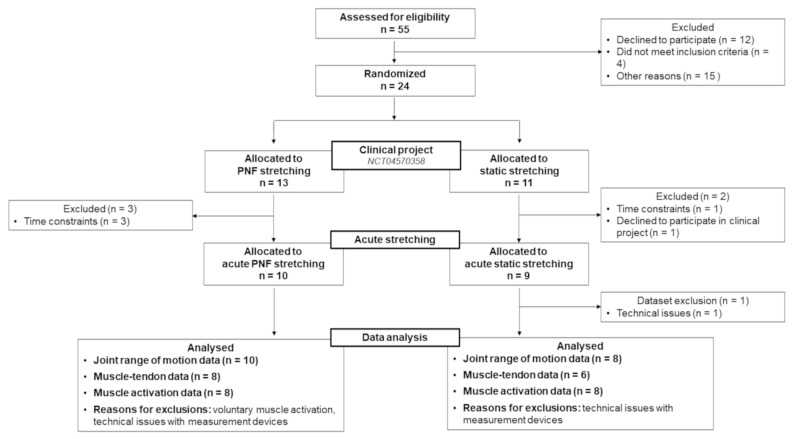
Flow diagram for the participants included in the present study.

**Table 1 ijerph-19-11599-t001:** Participant characteristics of the children with spastic cerebral palsy in the static stretching (SS) and proprioceptive neuromuscular facilitation (PNF) groups. Data are presented as mean (SD).

Anthropometrics	SS	PNF
Number	8	10
Gender (female/male)	5/3	3/7
Age (years)	10.8 (2.1)	11.3 (2.3)
Body mass (kg)	44.9 (13.2)	47.8 (16.5)
Body height (cm)	148.3 (9.5)	149.1 (16.8)
Lower leg length (cm)	36.1 (3.0)	35.4 (4.2)
**Patient characteristics**		
GMFCS level I/II/III	7/1/0	7/1/2
Affected (unilateral/bilateral)	3/5	4/6
SCALE (points: 2/1/0)	1/5/2	2/8/0

The SCALE grading is presented for the more affected leg and the affected leg in children with bilateral and unilateral cerebral palsy, respectively. Grading was limited to the ankle joint. Normal: 2 points; impaired: 1 point; unable: 0 points.

**Table 2 ijerph-19-11599-t002:** Ankle joint range of motion, and peak as well as mean muscle activation measured during dorsiflexion rotations before (pre) and after (post) both acute proprioceptive neuromuscular facilitation (PNF) stretching and acute static stretching (SS) in children with spastic cerebral palsy. Data are presented as mean (SD).

Parameter	Group	*n*	Pre	Post	%Δ_Pre-Post_	Effect	F	*p*	Partial η²
**Range of motion** (deg)	PNF	10	51.1 (8.2)	52.2 (9.0)	2.2	**Time**	**7.261**	**0.016 ***	**0.312**
	SS	8	48.3 (3.0)	51.9 (2.9)	7.4	Time × group	1.980	0.179	0.110
**Maximum dorsiflexion** (deg)	PNF	10	−7.3 (10.3)	−4.0 (9.0)	n.a.	**Time**	**4.900**	**0.042 ***	**0.234**
	SS	8	−14.5 (8.0)	−10.4 (4.3)	n.a.	Time × group	0.051	0.824	0.003
Foot sole angle change_commonT_ (deg)	PNF	8	28.9 (6.7)	28.9 (5.5)	−0.3	Time	3.459	0.088	0.224
	SS	6	26.9 (1.8)	30.4 (3.3)	13.1	Time × group	3.798	0.075	0.240
Resting angle (deg)	PNF	10	−47.5 (9.4)	−45.5 (9.4)	4.2	Time	0.530	0.477	0.032
	SS	8	−52.2 (7.0)	−51.9 (5.4)	0.6	Time × group	0.276	0.606	0.171
Peak externally applied torque (Nm)	PNF	10	8.2 (0.9)	8.5 (0.8)	2.9	Time	1.517	0.236	0.087
	SS	8	8.8 (1.1)	9.2 (1.0)	4.0	Time × group	0.060	0.809	0.004
Muscle activation (% of MVC) ^§^	PNF	8	2.3 (3.0)	2.0 (2.3)	−10.8	Time	0.968	0.342	0.065
	SS	8	0.9 (0.6)	0.7 (0.4)	−15.2	Time × group	0.058	0.813	0.004
Muscle activation_commonT_ (% of MVC) ^§^	PNF	8	2.2 (2.8)	2.1 (2.3)	−4.4	Time	0.087	0.772	0.006
	SS	8	0.8 (0.6)	0.7 (0.4)	−16.5	Time × group	0.141	0.713	0.01

* Significant effect (*p* < 0.05). ^§^ Variables were logarithmically transformed for analysis, but are displayed with the original mean and standard deviation, for comprehensibility. Negative angle values indicate plantarflexion; n.a., not applicable; MVC, maximum voluntary contraction; commonT, parameter calculated over a common moment interval ranging from 0 Nm to 5 Nm externally applied torque.

**Table 3 ijerph-19-11599-t003:** Gastrocnemius muscle-tendon unit length, muscle belly length, and Achilles tendon length changes, as well as strain values measured during dorsiflexion rotations before (pre) and after (post) both acute proprioceptive neuromuscular facilitation (PNF) stretching and acute static stretching (SS) in children with spastic cerebral palsy. Data are presented as mean (SD).

Parameter	Group	*n*	Pre	Post	%Δ_Pre-Post_	Effect	F	*p*	Partial η²
**Muscle-tendon unit elongation_commonT_ (mm)**	PNF	8	20.3 (4.7)	18.5 (3.1)	−8.8	Time	0.191	0.670	0.016
	SS	6	18.3 (2.5)	20.9 (4.6)	14.6	**Time × group**	**4.768**	**0.050 ***	**0.284**
Maximal muscle belly elongation (mm)	PNF	9	17.7 (6.1)	16.9 (6.5)	−4.1	Time	0.484	0.498	0.033
	SS	7	17.6 (6.2)	16.9 (9.3)	−3.5	Time × group	0.003	0.956	0.000
Muscle belly elongation_commonT_ (mm)	PNF	8	13.6 (5.6)	12.5 (5.7)	−7.5	Time	0.523	0.484	0.042
	SS	6	12.9 (3.7)	12.9 (6.0)	0.2	Time × group	0.569	0.465	0.045
Muscle belly strain_commonT_ (%)	PNF	8	7.9 (3.9)	7.2 (3.7)	−9.2	Time	0.606	0.451	0.048
	SS	6	7.0 (1.3)	7.0 (2.7)	1.2	Time × group	0.968	0.345	0.075
Tendon elongation_commonT_ (mm)	PNF	8	6.7 (3.7)	6.0 (3.6)	−11.3	Time	1.326	0.272	0.099
	SS	6	5.4 (2.5)	8.0 (2.8)	49.1	Time × group	4.308	0.060	0.264
Tendon strain_commonT_ (%)	PNF	8	3.9 (2.2)	3.4 (2.1)	−12.4	Time	1.099	0.315	0.084
	SS	6	3.4 (1.6)	5.0 (1.4)	45.9	Time × group	3.912	0.071	0.246

* Significant effect (*p* < 0.05). commonT, parameter measured over a common moment interval ranging from 0 Nm to 5 Nm externally applied torque.

## Data Availability

The data that support the findings of this study are available from the corresponding author [A.K.], upon reasonable request.

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
