# Peer review of "Acute Effects of Static and Proprioceptive Neuromuscular Facilitation Stretching of the Plantar Flexors on Ankle Range of Motion and Muscle-Tendon Behavior in Children with Spastic Cerebral Palsy—A Randomized Clinical Trial"

_ijerph, 2022, doi:10.3390/ijerph191811599_

Round 1

Reviewer 1 Report

This is an interesting study. The study addresses stretching to help in preventing muscle contracture in children with spastic cerebral palsy. Therapies  need to be assessed for clinical relevance and this is what this study is focused on.

The study is well presented and detailed. The figures in the type of movements are well documented and the assessments are thorough. The tables and results presented are informative as well as important for other researchers to have access to in the publication. The authors are careful not to over state their findings and have commented on their samples size being  limitation for some of the findings. While other results are significant others are trending to be significant and likely with a larger sample size more informative assessments will be able to be made.

One comment:

The 1st line The authors state CP is a “brain injury” but I am just curious if the brain develops this way  in development is it an “injury” or a “developmental pathology”. Maybe this is the way the field describes CP but just wanting to make sure “injury” is the correct description.

As a side note, but not needing to be addressed in the manuscript and no reply is needed. Just curiosity, would possibly a muscle massage (vibration) help in loosening the muscles prior to stretching and maybe off set the muscle spindles so that there may be less resistance feedback to passive movements for the stretching ? Possible muscle massage alone might help with reducing the spasticity associated with CP.

Reviewer 2 Report

The paper by Kruse et al. “Acute effects of static and proprioceptive neuromuscular facilitation stretching of the plantar flexors on ankle range of motion and muscle-tendon behavior in children with spastic cerebral palsy” describes an attempt to compare the effects of proprioceptive neuromuscular facilitation stretching and static stretching in CP children.

The investigation of this question represents an enrichment of the knowledge about possible forms of therapy for CP children. The results of the therapy by Kruse et al. show a positive effect of stretching on ankle function, with SS resulting in increased MTU elongation, while it decreased after PNF.

 Major:

A major problem lies in the study design. There is no control group! Also, with the relatively small n, no matching of the groups was carried out on a criterion that would have been relatively easy to use in the clinical assessment of spasticity.

A sample size calculation would certainly have been easily possible based on large amount of data available on static stretching.

There are also terminological problems: muscle belly extensibility or muscle belly excursion. What exactly does that mean? One can only speculate here, in any case, these formulations are imprecise and undefined.

 Line 149 ff Why were only two stretching exercises carried out, isn't there a risk that an "incorrect measurement" with a large weight will be included in the data?

It is also not clear in the text which results from the US sample delivers. Perhaps about at Line 195 one can include two images in the text. Why is the US data recorded with a 60 Hz frame rate? Are those the kinemetric sensors?

The data acquisition technique (hardware & software) is not described for this experiment. The selection of a 6th order Butterworth filter is not justified and cannot be derived from the SENIAN recommendations.

The presentation of the method should be revised significantly.

In the discussion, when the results are weighed up, there is clearly no reference to a necessary control group. It is not possible to reasonably assess the importance of the applied forms of therapy. The investigation with a control group, which is clinically matched to the treatment groups, must be carried out for the publication of this experiment.

minor:

Line 91 white

Line 160 better “measured” than “assessed”

Line 170 ff Placement of reflective markers, ultrasound transducer, and electromyographic sensors to assess foot sole angles, gastrocnemius medialis muscle belly behavior, and muscle activity throughout dorsiflexion rotations, respectively.

The sentence should be rewritten, when I first read it I thought, which means: "electromyographic sensors to assess foot sole angles".

Line 187 and FF In the heading, abbreviations do not contribute to readability

Reviewer 3 Report

Dear Authors,

I read your manuscript with interest since I believe it deals with a relevant topic. Overall, the methodology is strong and the rationale is sufficiently elaborated, although it can be improved with an attempt of describing more in details the differences between the two techniques from a physiological perspective referring to children with CP. 

I have one major concern about the sample size, since in the registered protocol you calculated a sample size of 30 subjects but here you randomized only 24 subjects (further reduced in the analyses). I am concerned that this may affect the statistical power of the analyses. To rule out this hypothesis I suggest recruiting some more patients to reach the original estimated sample size.

A few minor suggestions: 

- add "randomized clinical trial" to the title and in the methods sections according to CONSORT checklist.

- please clarify if these are re results from a single session of SS/PNF or the results of an 8-week program s stated in the trial protocol.

- the methods section is rather long and with too many subheadings hampering readability.

- it is not clear to me why you used an ANOVA approach rather than a unpaired t-test

- some data presented in the methods section should be moved to the results (clinical characteristics of the sample, consort flowchart..)

- please elaborate a bit more on the significance of improving ankle dorsiflexion ROM of 3 degrees.

- Do you have any hypothesis on why SS seems to be more effective than PNF in these subjects in terms of muscle-tendon unit elongation and ROM?

Overall I suggest major revision.
